# Quantum nondemolition measurement of mechanical motion quanta

Luca Dellantonio[1,2], Oleksandr Kyriienko [1,3], Florian Marquardt [4,5] & Anders S. Sørensen [1,2]

The fields of optomechanics and electromechanics have facilitated numerous advances in the areas of precision measurement and sensing, ultimately driving the studies of mechanical systems into the quantum regime. To date, however, the quantization of the mechanical motion and the associated quantum jumps between phonon states remains elusive. For optomechanical systems, the coupling to the environment was shown to make the detection of the mechanical mode occupation difficult, typically requiring the single-photon strong-coupling regime. Here, we propose and analyse an electromechanical setup, which allows us to overcome this limitation and resolve the energy levels of a mechanical oscillator. We found that the heating of the membrane, caused by the interaction with the environment and unwanted couplings, can be suppressed for carefully designed electromechanical systems. The results suggest that phonon number measurement is within reach for modern electromechanical setups.

[1] The Niels Bohr Institute, University of Copenhagen, Blegdamsvej 17, 2100 Copenhagen Ø, Denmark. [2] Center for Hybrid Quantum Networks (Hy-Q), The Niels Bohr Institute, University of Copenhagen, Blegdamsvej 17, 2100 Copenhagen Ø, Denmark. [3] NORDITA, KTH Royal Institute of Technology and Stockholm University, Roslagstullsbacken 23, 106 91 Stockholm, Sweden. [4] Institute for Theoretical Physics, University Erlangen-Nürnberg, Staudstraße 7, 91058 Erlangen, Germany. [5] Max Planck Institute for the Science of Light, Günther-Scharowsky-Straße 1, 91058 Erlangen, Germany. Correspondence and requests for materials should be addressed to L.D. (email: luca.delantonio@nbi.ku.dk)

Energy quantization is one of the hallmarks of quantum mechanics. First theorized for light by Einstein and Planck, it was found to be ubiquitous in nature and represents a cornerstone of modern physics. It has been observed in various microscopic systems starting from nuclei, atoms and molecules, to larger mesoscopic condensed matter systems such as super-conductors[1]. For macroscopic systems, however, the observation of energy quantization is hindered by the smallness of the Planck constant. Thus, although being a milestone of contemporary physics, up to date the discrete energy spectrum of mechanical resonators has never been seen directly.

Extreme progress in studying mechanical systems has been achieved in experiments exploiting radiation pressure. This is the core of optomechanics[2], where photons and phonons of the optical and mechanical subsystems interact with each other. A similar type of coupling can be realized in the microwave domain with electrical circuits, leading to the field of electromechanics[3–8]. The numerous advances of optomechanics and electromechanics include ground state cooling[4,5,9–11], ultra precise sensing[12–15], generation of squeezed light and mechanical states[7,8,16–18], back action cancellation[19,20] and detection of gravitational waves[21]. In all of these systems, however, the operation in the single-photon/phonon regime is challenging due to the small value of the bare coupling[3,22]. Instead, experiments exploit an enhanced linearized effective coupling induced by a large driving field. This severely limits the nature of the interactions[23] and possible quantum effects. In particular, it precludes the observation of the energy quantization in mechanical resonators.

Quantization of mechanical energy can be observed by a quantum nondemolition (QND) measurement[24,25] of an oscillator's phonon number operator $\hat{n}_b$. Here, QND means that the interaction, which couples the mechanical system with the measurement apparatus, does not affect the observable we are interested in. This is achieved if the total Hamiltonian commutes with $\hat{n}_b$, and the influence of the environment is minimized.

Considering the electromechanical setups in Fig. 1, we show that QND detection is feasible for a capacitor in which one of the electrodes is a light micromechanical oscillator. By choosing an antisymmetric mode for the oscillator, the interaction between the electrical and mechanical subsystems is quadratic in the displacement. Along with the suppression of the linear coupling, this ensures the QND nature of the measurement, as originally proposed in refs. [26,27] for an optomechanical system. In that system, however, it was shown in refs. [28,29] that the combination of unwanted losses and the coupling to an orthogonal electromagnetic mode spoils the interaction, unless strong single-photon coupling is achieved. Here, we show that for the considered electromechanical setup the equivalent orthogonal mode can have dramatically different properties, allowing for the phonon QND detection. We derive general conditions under which the QND measurement is possible, and characterize its experimental signatures. As compared to most approaches to phonon QND measurement[26,27,30–32], our procedure does not impose stringent requirements on the single-photon optomechanical coupling, but relies on the ratio of the involved coupling constants. This makes our approach attractive even for systems where the interaction is limited, for example, due to stray capacitances in the setup. For a measurement of the square displacement, a similar advantage was identified in ref. [31].

## Results

**Proceeding.** We first study an *RLC* circuit with one capacitor plate being an oscillating membrane, without assuming the symmetry discussed above (Fig. 1b). The mechanical motion of the plate shifts the resonance frequency of the circuit, while the

electric potential exerts a force on the membrane. In order to perform a QND measurement of the phonon number, we require this interaction to be proportional to $\hat{n}_b$. We therefore Taylor expand the inverse of the capacitance to second order in the displacement, $1/C(\hat{x}) \simeq C_0^{-1} + \tilde{g}_1(\hat{b} + \hat{b}^\dagger) + \tilde{g}_2(\hat{b} + \hat{b}^\dagger)^2/2$, where we replaced the position $\hat{x}$ with the creation $\hat{b}^\dagger$ and annihilation $\hat{b}$ operators of the mechanical motion, and $\tilde{g}_{1,2}$ denote linear and quadratic coupling constants. Within the rotating wave approximation, $\tilde{g}_2(\hat{b} + \hat{b}^\dagger)^2/2 \simeq \tilde{g}_2\hat{n}_b$, leading to the desired QND interaction, while the $\tilde{g}_1$ term adds unwanted heating that spoils the phonon measurement.

The main aim of this work is to identify conditions under which the QND measurement is feasible, despite the presence of heating. We initially consider the simple circuit in Fig. 1b, and assume the incoming signal $\hat{V}_{in}$ to be in a coherent state resonant with the circuit. The quadratic interaction then shifts the electrical resonance frequency proportionally to the phonon number $\tilde{g}_2\hat{n}_b$. For small $\tilde{g}_2$, this shift leads to a phase change of the outgoing signal $\hat{V}_{out}$ that can be determined by homodyne measurement. Different phononic states will thus lead to distinct outcomes $V_M$, as shown in Fig. 2. The distance $d$ between output signals for different $\hat{n}_b$ and the standard deviation $\sigma$ of the noise define the signal-to-noise ratio $D = d/\sigma$ (see Fig. 2), which needs to be maximized.

In order to have a successful QND measurement, the phonon number $\hat{n}_b$ must be conserved. If the mechanical state jumps during a measurement, the outcome $V_M$ ends up between the desired peaks. This leads to a reduced contrast, as illustrated by the distribution in the background of Fig. 2. The probability for $\hat{n}_b$ to change is generally state-dependent, in the sense that higher Fock states are more likely to jump. A state-independent characterization of this heating is given by the average phonons $\Delta n_b$ added to the ground state during the measurement time $T$. The jump probability for any state can then be derived from $\Delta n_b$ using standard results for harmonic oscillators (for details see Supplementary Note 3 available in Supplementary Material online).

Both $D$ and $\Delta n_b$ are proportional to the incoming intensity. We therefore characterize a setup by the parameter $\lambda = D^2/\Delta n_b$, where $\lambda \gg 1$ is required for successful QND detection. For the *RLC* circuit in Fig. 1b, we find below that

$$\lambda = \frac{1}{2(1 + 2\bar{n}_e)^2} \left(\frac{g_2}{g_1}\right)^2 \left(\frac{\omega_m}{\gamma_t}\right)^2, \tag{1}$$

where $g_1 = \tilde{g}_1 C_0 \omega_s$, $g_2 = \tilde{g}_2 C_0 \omega_s$ and $\bar{n}_e$ is the thermal occupation of $R_0$ and $Z_{out}$ (assumed equal, $R_0 = Z_{out}$). Here, $\omega_m$ and $\omega_s = (C_0 L_0)^{-1/2} \gg \omega_m$ are the mechanical and electrical frequencies, respectively, and $\gamma_t = Z_{out}/L_0$ corresponds to the output coupling rate. A result similar to Eq. (1) is derived in ref. [33].

Despite progresses in reaching the resolved sideband regime $\omega_m \gg \gamma_t$ in both optomechanical and electromechanical systems, $g_2$ is generally much smaller than $g_1$, implying $\lambda \ll 1$ in Eq. (1). To circumvent this problem, we use the second fundamental mode of the membrane in the capacitor, as depicted in Fig. 1a. The first-order coefficient $\tilde{g}_1$ of the $1/C(\hat{x})$ expansion then vanishes, leaving $\tilde{g}_2$ to be the largest contribution to the electromechanical coupling. In this situation $\lambda$ seemingly grows indefinitely, the induced heating disappears and the QND measurement of the phonon number is easily realized. In practice, however, two effects will limit the achievable value of $\lambda$. First, inaccuracies in the nanofabrication can cause misalignments and, consequently, a residual linear coupling. Second, the oscillation of the membrane induces a charge redistribution in the capacitor to maintain it at an equipotential. The associated antisymmetric electrical mode introduces an effective linear coupling, and a similar heating mechanism as the one identified

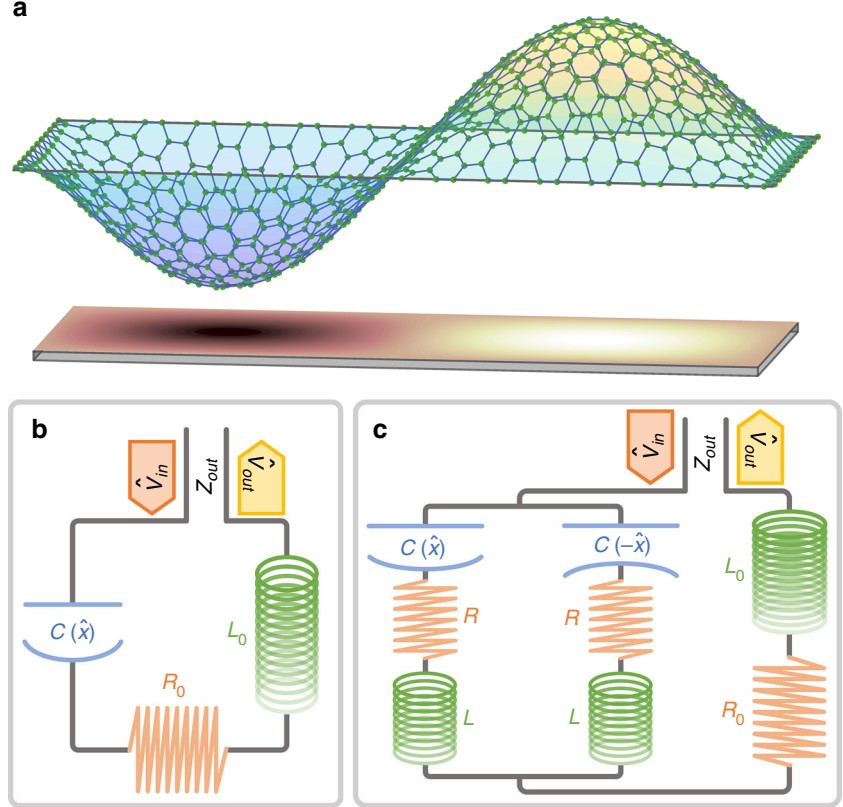

**Fig. 1** Experimental setup. **a** Sketch of a capacitor with an oscillating plate, here represented by a graphene membrane. We consider an antisymmetric (2, 1) mechanical mode. **b** RLC oscillator formed by the inductance $L_0$, resistance $R_0$, and position-dependent capacitance $C(\hat{x})$. The circuit is driven by the input voltage $\hat{V}_{in}$ through a transmission line of impedance $Z_{out}$. $\hat{V}_{out}$ is the reflected signal. **c** Model circuit for an RLC system where the capacitor has the same form as in **a**. The membrane has a vanishing linear coupling to the symmetric electrical mode used for probing the system. The antisymmetric mode, residing in the small loop containing parasitic inductances $L$ and resistances $R$, describes the redistribution of charge on the capacitor

in ref.[28] for the optomechanical setup of refs[26,27]. In these papers, the quadratic interaction results from a hybridization of two modes linearly coupled to the mechanical position, and the QND detection was found to be impossible unless the single-photon coupling $g_1$ exceeded the intrinsic cavity damping. In our case, the QND interaction arises directly from the Taylor expansion of the capacitance. Hence, there is no constraint tying the second-order coupling $g_2$ to the properties of the symmetric and antisymmetric electrical modes, which can have vastly different resonance frequencies and dampings. This inhibits the mechanical heating and ultimately allows for the QND detection of the phonon number. We model the charge redistribution in the capacitor by parasitic inductances ($L$) and resistances ($R$) in the equivalent circuit of Fig. 1c. Each of the two arms containing $R$ and $L$ represents one half of the capacitor, with opposite dependence on the membrane position, $C(\hat{x})$ and $C(-\hat{x})$.

**Single-arm RLC circuit**. In the following, we derive Eq. (1) for the RLC circuit in Fig. 1b. The methods sketched here will then be generalised for the double-arm circuit in Fig. 1c. Using the standard approach[34], we write the circuit Hamiltonian as $\hat{\mathcal{H}}(\hat{x}) = \hat{\Phi}^2/[2L_0] + \hat{Q}^2/[2C(\hat{x})]$, where the conjugate variables $\hat{Q}$ and $\hat{\Phi}$ are the charge and magnetic flux, respectively. We can expand $\hat{\mathcal{H}}(\hat{x})$ in the mechanical position $\hat{x} \propto \hat{b} + \hat{b}^\dagger$, in order to obtain the circuit Hamiltonian $\hat{\mathcal{H}}_e = \hat{\mathcal{H}}(\hat{x} = 0)$ and the coupling Hamiltonian $\hat{\mathcal{H}}_{em} = g_1 \omega_s L_0 \hat{Q}^2(\hat{b} + \hat{b}^\dagger)/2 + g_2 \omega_s L_0 \hat{Q}^2(\hat{n}_b + \hat{b}\hat{b}/2 + \hat{b}^\dagger\hat{b}^\dagger/2)$. The total Hamiltonian $\hat{\mathcal{H}}_{tot} = \hat{\mathcal{H}}_e + \hat{\mathcal{H}}_{em} + \hat{\mathcal{H}}_m$ is therefore the sum of the circuit, interaction and the mechanical Hamiltonian $\hat{\mathcal{H}}_m = \hbar\omega_m \hat{b}^\dagger \hat{b}$.

Next, we describe the environmental effects corresponding to decay and heating of the modes. Associating each resistor $R_i$ with its own Johnson–Nyquist noise $\hat{V}_{R_i}$, we find the equations of motion of the composite system

$$\dot{\hat{Q}} = \frac{\hat{\Phi}}{L_0}, \qquad (2)$$

$$\dot{\hat{\Phi}} = -\frac{\hat{Q}}{C_0} - (\gamma_t + \gamma_r)\hat{\Phi} - g_1 \omega_s L_0 \hat{Q}(\hat{b} + \hat{b}^\dagger)$$
$$- g_2 \omega_s L_0 \hat{Q}\left(\hat{n}_b + \frac{\hat{b}\hat{b} + \hat{b}^\dagger\hat{b}^\dagger}{2}\right) + 2\left(\hat{V}_{in} + \hat{V}_{R_0}\right), \qquad (3)$$

$$\dot{\hat{b}} = -i\omega_m \hat{b} - g_1 \frac{i\omega_s L_0 \hat{Q}^2}{2\hbar} - g_2 \frac{i\omega_s L_0 \hat{Q}^2}{2\hbar}\left(\hat{b} + \hat{b}^\dagger\right) - \frac{\gamma_b}{2}\hat{b} + i\frac{x_0}{\hbar}\hat{F}_b, \qquad (4)$$

where $\gamma_r = R_0/L_0$, $\gamma_b$ is the intrinsic mechanical damping rate with associated noise $\hat{F}_b$ and $x_0 = \sqrt{\hbar/(2m\omega_m)}$ is the amplitude of the zero-point motion for a membrane of mass $m$. From now on, we consider optimally loaded setups with $\gamma_r = \gamma_t$. Equations (2)–(4) fully characterize the dynamics of the system, and represent the starting point for our detailed analysis.

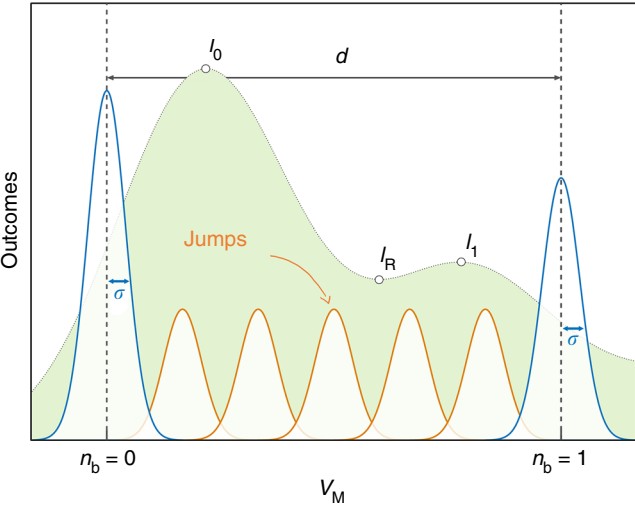

**Fig. 2** Sketch of the experimental outcome. Distribution of outcomes $V_M$ for two different phonon numbers: $n_b = 0$ (first peak to the left) and $n_b = 1$ (last peak to the right). For a given value of $n_b$, repeated measurements are Gaussian distributed with a variance $\sigma^2 \propto 1 + 2\bar{n}_e$ of the outgoing signal $\hat{V}_{\text{out}}$, consisting of vacuum and thermal noise. The distance $d$ between the two peaks depends on the circuit parameters and the number of incident photons, and identifies the signal-to-noise ratio $D = d/\sigma$. Ideally, for each shot of the measurement, the mechanics is either in its ground or first excited state. However, for $\Delta n_b > 0$ there will be events where the mechanical state jumps, resulting in outcomes $V_M$ in between the peaks relative to $n_b = 0$ and $n_b = 1$ (smaller peaks in the figure). This leads to the smeared distribution shown in the back. The visibility of the QND measurement is quantified by the values at the peaks and valleys, as indicated by $I_0$, $I_1$ and $I_R$ (see Eq. (9)). The figure is for illustration only, and is not to scale

The feedback of the membrane's motion on the electrical circuit is described by Eq. (3). Driving the system at the electrical resonance frequency $\omega_s$, the terms proportional to $g_1(\hat{b} + \hat{b}^\dagger)$ and $g_2(\hat{b}\hat{b} + \hat{b}^\dagger\hat{b}^\dagger)$ give rise to sidebands at frequencies $\omega_s \pm \omega_m$ and $\omega_s \pm 2\omega_m$, respectively, whereas $g_2\hat{n}_b$ induces a phonon-dependent frequency shift of the microwave cavity. Since homodyne detection is only sensitive to signals at the measured frequency, the sidebands are removed in the outcome $V_M$, which is defined as the phase quadrature of $\hat{V}_{\text{out}} = \hat{V}_{\text{in}} - \gamma_t\hat{\Phi}$. This allows us to neglect oscillating terms in the calculation of $V_M$ (the linear term also leads to mechanically induced damping, but this is typically negligible compared to $\gamma_t$). The only contribution to $V_M$ is therefore the phonon-dependent frequency shift, which allows us to resolve the mechanical state. On the contrary, the electrically induced mechanical heating only involves the sidebands $\omega_s \pm \omega_m$ and $\omega_s \pm 2\omega_m$, being unaffected by the term $g_2\hat{n}_b$ in the Hamiltonian. For the RLC circuit in Fig. 1b, the heating is dominated by the linear term, since $g_1 \gg g_2$, and we shall neglect $g_2$ for the calculation of $\Delta n_b$.

Below, we quantify the heating of the membrane and the phonon-dependent LC frequency shift. We first assume that the mechanical state does not jump during the measurement. Then, the equations of motion of the two subsystems decouple and we find $D^2 = g_2^2|\alpha|^2/[4(1 + 2\bar{n}_e)\gamma_t^2]$, where the number of photons $|\alpha|^2$ sent into the circuit within the measurement time $T$ sets the measurement strength. As discussed above, $\Delta n_b$ is the average phonon number at the end of the measurement $\Delta n_b = \langle \hat{n}_b(T) \rangle$, with the mechanics initially in its ground state. For $T$ much shorter than the mechanical lifetime $\gamma_b^{-1}$, $\Delta n_b$ can be linearized to find the rate at which the membrane heats up. For the RLC circuit in Fig. 1b, we find $\Delta n_b = (1 + 2\bar{n}_e)g_1^2|\alpha|^2/(2\omega_m^2)$. The parameter $\lambda$ given in Eq. (1) is then found as the ratio $\lambda = D^2/\Delta n_b$. For details see Supplementary Note 1 available in Supplementary Material online.

**Double-arm circuit**. With the overall linear coupling vanishing, the parameter $\lambda$ will be limited by fabrication imperfections and coupling to the antisymmetric mode. To model these phenomena,

we consider the circuit in Fig. 1c, where the antisymmetric mode resides inside the small loop containing the two capacitors, and the symmetric one probes the system. We derive $g_1$ and $g_2$ from the expansion of each of the two capacitors: $1/C(\pm\hat{x}) \simeq C_0^{-1} \pm \tilde{g}_1(\hat{b} + \hat{b}^\dagger) + \tilde{g}_2\hat{n}_b$, so that in the absence of fabrication imperfections the total capacitor $C_{\text{tot}} = C(\hat{x}) + C(-\hat{x})$ is not linearly coupled to the symmetric mode. The coefficients $g_1$ and $g_2$ are related to their tilde counterparts in the same way as before, and the parameters $D^2$ and $\Delta n_b$ are evaluated in a similar fashion as we did for the RLC circuit. Since we quantify two sources of heating, it is convenient to write $\lambda = (\lambda_b^{-1} + \lambda_p^{-1})^{-1}$, where $\lambda_b$ takes into account heating from charge redistribution, and $\lambda_p$ describes the influence of fabrication imperfections. With the details presented in Supplementary Note 2 (for details, see Supplementary Material available online) and Methods, we find

$$\lambda_b = \frac{2}{(1 + 2\bar{n}_e)^2}\left(\frac{g_2}{g_1}\right)^2\left(\frac{\omega_s}{\gamma_t}\right)^2\frac{Z_{\text{out}}}{R}, \qquad (5)$$

$$\lambda_p = \frac{2}{(1 + 2\bar{n}_e)^2}\left(\frac{g_2}{g_1}\right)^2\left(\frac{g_1}{g_r}\right)^2\left(\frac{\omega_m}{\gamma_t}\right)^2, \qquad (6)$$

where $\omega_s = [C_0(L + 2L_0)]^{-1/2}$ is the frequency of the symmetric mode, $\gamma_t = [2Z_{\text{out}}]/[L + 2L_0]$ is the decay to the transmission line and $g_r = 2C_0x_0\omega_s\partial_xC_{\text{tot}}^{-1}(x)$ is the residual linear coupling induced by fabrication imperfection. We use the same notation introduced for the RLC circuit to allow a direct comparison. Equations (5) and (6) express the gain of our approach to QND detection. First, Eq. (6) quantifies the advantage of symmetry: $\lambda$ dramatically improves compared to Eq. (1) by having a small residual linear coupling $g_r \ll g_1$. Second, Eq. (5) is multiplied by the factor $(\omega_s/\omega_m)^2$ with respect to Eq. (1). For microwave readout of a megahertz oscillator, this factor can be substantial. Furthermore, the mechanical oscillator is now only susceptible to the noise associated with charge redistribution on the capacitor, and not to the resistance in the inductor. This gives an additional improvement if $R < Z_{\text{out}}$.

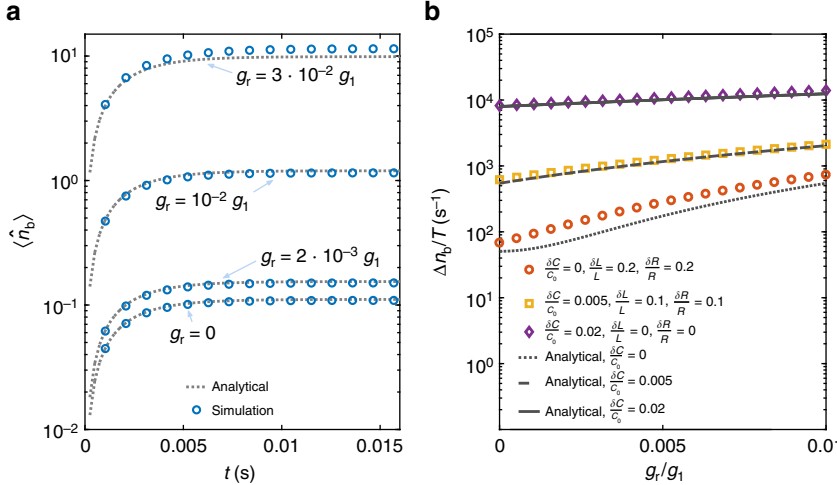

**Fig. 3** Heating simulations. **a** Average phonon number $n_b(t)$ as a function of time. We present a comparison between the analytical curves (grey, dotted lines) and the full simulations of the system (blue dots). From the bottom to the top we set $g_r/g_1$ to be 0, $2 \times 10^{-3}$, $10^{-2}$ and $3 \times 10^{-2}$. We use $\delta R = \delta L = \delta C = 0$. **b** Heating rate $\Delta n_b/T$ as a function of the normalized residual linear coupling $g_r/g_1$. Here we analyse the system in the presence of asymmetries in the parasitic elements of the circuit. The three dark grey lines are the analytical predictions for $\delta C/C_0$ being equal to 0 (dotted), 0.005 (dashed) and 0.02 (full). The circles, squares and diamonds are the simulated results for the values $\delta R/R$, $\delta L/L$ and $\delta C/C$ reported in the legend. We assume $L/L_0 = 10^{-2}$, $R/Z_{out} = 10^{-1}$, $\omega_s = (2\pi)7$ GHz, $\omega_m = (2\pi)80$ MHz, $\gamma_r \simeq \gamma_t = (2\pi)0.15$ MHz, $\gamma_b = (2\pi)80$ Hz, $g_1 = (2\pi)7$ kHz, $\bar{n}_e = \bar{n}_m = 0$, and an incident photon flux $|\tilde{\alpha}|^2 = 1.15 \times 10^{15}$ s$^{-1}$

To describe a realistic situation, we numerically simulate the case in which the parasitic resistances $R$, inductances $L$ and the two bare capacitances $C_0$ differ from each other. In Fig. 3, we test the system with these asymmetries and the physical parameters given below. In the left plot, the role of a residual linear coupling $g_r$ is investigated. In the right one, we consider unbalanced resistances $R \pm \delta R$, inductances $L \pm \delta L$ and capacitances $C_0 \pm \delta C$. The results show that our analytical predictions accurately describe a system with non-zero $g_r$ and $\delta C$. Furthermore, the numerical points confirm that $\delta R$ and $\delta L$ enter as higher order perturbations. In fact, we generally find that Eqs. (5) and (6) are accurate for relatively large perturbations (up to 25%).

Inspired by recent experiments[15,35–38], we estimate the value of $\lambda$, which can be reached in state-of-the-art setups. We consider a rectangular monolayer graphene membrane of length 1 μm and width 0.3 μm, with a mechanical frequency of $\omega_m = (2\pi)80$ MHz and a quality factor $Q = 10^6$. It is suspended $d_0 = 10$ nm above a conducting plate, forming the capacitor (see sketch in Fig. 1a). Assuming that the membrane is clamped to the substrate along its boundaries, we identify the ratio of the coupling coefficients for each capacitor $C(\pm \hat{x})$ in Fig. 1c to be $g_2/g_1 = \pi^2 x_0/(8 d_0)$[39]. Considering that for these geometries stray capacitances $C_s$ are typically preponderant with respect to $C_0$, we take $g_1 \simeq (2\pi)7$ kHz and $g_2 \simeq (2\pi)1$ Hz, corresponding to $C_s \simeq 100 C_0$. For comparison, a value of $C_s = 50$ fF is obtained in ref. [35], for a graphene membrane about two and a half times the size considered here. This stray capacitance would be 376 times $C_0 \simeq 13$ fF. Assuming a reduction of $C_s$ due to the smaller dimensions, we take $C_s = 100 C_0$.

With an electrical reservoir at zero temperature $\bar{n}_e \simeq 0$ (valid for milliKelvin experiments), an electrical frequency $\omega_s = (2\pi)7$ GHz and decay rate $\gamma_t = (2\pi)150$ kHz, we get $\lambda_b = 105 \times Z_{out}/R$ and $\lambda_p = 0.014 \times (g_1/g_r)^2$. Since the graphene coupling can be tuned via electric fields[40–42], we assume $g_1/g_r \sim 100$, which fixes $\lambda$ between 60 ($R = Z_{out}$) and 122 ($R = Z_{out}/10$), mostly restricted by $\lambda_p$. This limit is well above the threshold for having a good visibility of the phonon number states (see below), and can be further improved by either increasing the sideband resolution $\omega_m/\gamma_b$, the electrical frequency $\omega_s$ or by reducing the size of the

membrane. In Fig. 4b, we show the linear coupling $g_1$ as a function of the stray capacitance. For small values of $C_s$, we reach the strong-coupling regime, where $g_1 \geq \gamma_t$. In the realistic scenario described above, where $C_s \gg C_0$, our scheme still allows for phonon QND measurement even for $g_1, g_2 \ll \gamma_t$. This is in contrast to the typical optomechanical approach, where the quadratic interaction results from a hybridization of two optical modes, and strong coupling $g_1 > \gamma_t$ is required[28]. Regardless of how much $C_s$ reduces the coupling constants, it is in principle always possible to compensate by using stronger power. For details see the Supplementary Note 4 available in Supplementary Material online).

**Measurement**. We now evaluate how well a given value of $\lambda$ allows for the QND detection of the phonon number. To this end, we consider a situation where the system is continuously probed and measured. The output is then turned into discrete results by averaging over a suitable time $T$, and a histogram is constructed from the measured values $V_M$. We assume that the heating of the continuous QND probing is in equilibrium with the mechanical damping and the associated reservoir. In this case, one also needs to consider the thermal bath of the membrane. In addition to $\Delta n_b$ determined above, the total heating out of the ground state is thus $\Delta n_b + \gamma_b \bar{n}_m T$. This additional term leads to a redefinition of the parameter $\lambda$ to

$$\lambda' = \lambda \frac{\Delta n_b}{\Delta n_b + \gamma_b \bar{n}_m T}, \tag{7}$$

and the equilibrium average mechanical occupation, resulting from both the mechanical reservoir and the QND probe, becomes

$$\bar{N}_{eff} \simeq \bar{n}_m \frac{\lambda}{\lambda - \lambda'}. \tag{8}$$

The phonon QND measurement is then characterized by $\lambda'$, which is desirable to have as close as possible to its maximum $\lambda$. This can be achieved by choosing a sufficiently strong probing power and a short measurement time $T$, such that the mechanical heating can be neglected. This leads to a large $\bar{N}_{eff}$, which does

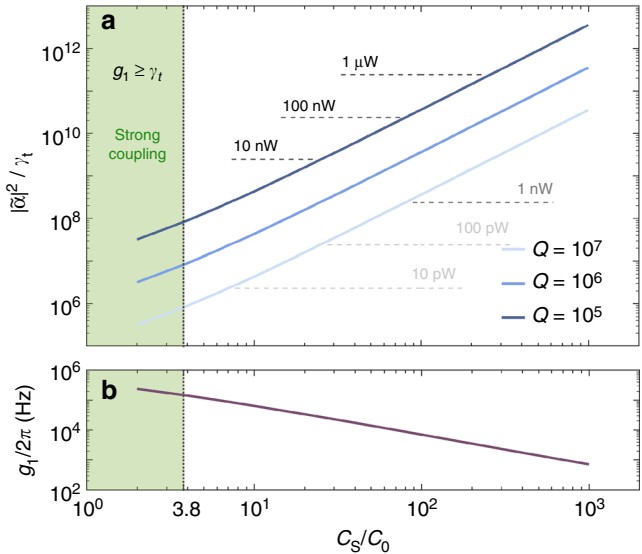

**Fig. 4** Operating conditions in the presence of stray capacitances. **a** Average intracavity photons $|\tilde{\alpha}|^2/\gamma_t$ required for the QND measurement, as a function of the relative value of the stray capacitance $C_s/C_0$. The three lines correspond to different values of the mechanical quality factor, as indicated in the legend. We assume $\Delta n_b = 0.3$ and equal contributions from the mechanical and electrically induced reservoirs $\bar{n}_m = \bar{N}_{eff}/2 = 3$. As a reference, the grey dashed lines indicate the associated powers of the probe. **b** Linear coupling $g_1$ as a function of $C_s/C_0$. For both figures, the shadowed region indicates the strong coupling $g_1 \geq \gamma_t$, where QND detection is feasible with other approaches[26, 31, 32]

not significantly change the contrast of the QND measurement (see Eq. (10) and Fig. 5b), but increases the time for acquiring significant statistics (the mechanical system spends less time in each Fock state).

Given $\lambda'$, we now want to optimize all remaining parameters of the system, to be able to discern the ground and first excited states with the largest contrast. We simulate the mechanical system with the quantum-jump method, and pick Gaussian distributed random values for the electrical vacuum and thermal noise. From this, we make the histogram of the resulting output voltages $V_M$ presented in Fig. 5a, where the induced heating $\Delta n_b$ is optimized numerically. For the optimization we consider the visibility

$$\xi = \frac{\frac{1}{2}(I_0 + I_1) - I_R}{\frac{1}{2}(I_0 + I_1) + I_R}, \quad (9)$$

where $I_0$ and $I_1$ are the heights of the peaks corresponding to $n_b = 0$ and $n_b = 1$ phonons, while $I_R$ is the lowest height in between $I_0$ and $I_1$ (see Fig. 2).

Additionally, we make an analytical model where we allow for one jump during each measurement period. We can extract the asymptotic behaviour of the visibility

$$\xi(\lambda', \bar{N}_{eff}) = 1 - 8\frac{3 + 5\bar{N}_{eff}}{1 + 2\bar{N}_{eff}}\frac{\sqrt{\pi \log \lambda'}}{\lambda'}, \quad (10)$$

reflecting the compromise between the contributions to $I_R$ from the noise $\propto \exp(-D^2/8)$ and from the jumps during the measurements $\propto \Delta n_b$.

The results of simulations and model are shown in Fig. 5a. The blue points are the numerical optimization, which are in good agreement with the analytical result (red, dotted line). Notice that for small values of $\lambda'$, the optimal $\Delta n_b$ is sufficiently high to allow multiple jumps during the measurement time $T$, leading to minor

discrepancies. The black, solid line is Eq. (10), and the shadowed region corresponds to the predicted values of $\lambda$ for the parameters introduced above. Qualitatively, clear signatures of the mechanical energy quantization are present for $\lambda' \gtrsim 40$, where the visibility exceeds 20%.

For the experimental parameters considered above, the maximum attainable value of $\lambda'$ is $\lambda = 122$ (for $R = Z_{out}/10$), and is achieved with a strong probe such that $\bar{N}_{eff} \gg \bar{n}_m$. The incident power and the measurement time $T$ provide a handle to optimize the performance for given experimental conditions. Qualitatively, a short value of $T$ minimizes the effects of the mechanical heating, and makes $\lambda' \simeq \lambda$. On the other hand, the required power to reach such a regime can be troublesome[43], and we may need to integrate for too long time to have sufficient statistics (since $\bar{N}_{eff} \gg 1$). This last problem can be solved by adding an electrical cooling, red-detuned by $\omega_m \gg \gamma_t$ from the QND probe. This cooling would not affect the parameter $\lambda'$, since it does not heat up the system, but only reduces $\bar{N}_{eff}$. The visibility $\xi$ thus remains almost unaltered (see Eq. (10) and Fig. 5b), but the probability to find the membrane in low excited states is increased, reducing the experimental time.

As an example, assume that the heating from the electrical feedback and the mechanical bath are equal, such that $\lambda' = \lambda/2 = 61$. Considering a cryogenic temperature of $14$ mK[37], the average mechanical occupation is $\bar{n}_m \simeq 3$, implying $\bar{N}_{eff} = 6$. The optimal $\Delta n_b$ is then 0.3, and can be obtained with a driving power of 16 nW and a measurement time of 0.1 ms for a mechanical quality factor $Q = 10^6$ and a stray capacitance $C_s = 100C_0$. For other values of $Q$ and $C_s$, the driving power can be varied to fulfil the constraint $\bar{N}_{eff} = 2\bar{n}_m$, as shown in Fig. 4a. The incident field is rather intense, which may cause additional heating to the system. In the set up of ref. [43], such additional heating has been observed above an intracavity photon number of $10^8$. For comparison, in Fig. 4a we show the intracavity photon number $|\tilde{\alpha}|^2/\gamma_t$ for our system, where $|\tilde{\alpha}|^2 = |\alpha|^2/T$ is the photon flux. Depending on the parameters, we see that $|\tilde{\alpha}|^2/\gamma_t$ will be similar or higher than $10^8$ for $C_s \gtrsim 100C_0$. These devices cannot, however, be compared directly. Nevertheless, since ref. [43] indicates that the source of this heating is electrical, we believe that it would be strongly suppressed for the QND measurement considered here. Since the linear coupling is almost cancelled by symmetry, the resulting heating rate is likely reduced by a factor $(g_r/g_1)^2 \simeq 10^{-4}$. In absence of this suppression, conducting our experiment in a pulsed regime may substantially reduce other heating mechanisms.

## Discussion

We have revisited the challenge of performing a phonon QND measurement. Employing symmetry to inhibit the linear coupling, the detrimental heating is suppressed while retaining the desired quadratic coupling. Contrary to the generally studied optomechanical case[28], the residual coupling to the antisymmetric mode is strongly suppressed by its higher frequency and reduced resistance. A particularly attractive feature of the current approach is that it is only sensitive to the ratio $g_2/g_1$, and not to their absolute values. Stray capacitances, which reduce the electromechanical couplings, can thus be compensated using stronger input fields.

These attractive features put QND detection within reach of presently available technology. A successful realization of a QND detection will not only represent a demonstration of genuine non-classical behaviour of mechanical systems, but also extend the interactions available in electro/optomechanics to non-Gaussian operations[44]. This will considerably expand the realm of effects that can be studied with these systems, and facilitate their application for quantum information processing[23].

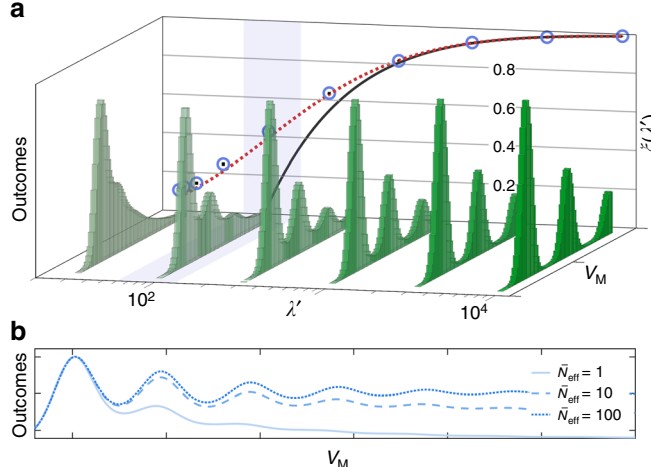

**Fig. 5** Visibility of the phonon QND measurement. **a** 3D plot: histograms of outcomes for different $\lambda'$ (from left to right, $\lambda' = 32$, $10^2$, $3 \times 10^2$, $10^3$, $3 \times 10^3$ and $10^4$). The optimal values of $\Delta n_b$ are (from left to right) 0.43, 0.27, 0.12, 0.05, $2 \times 10^{-3}$ and $8 \times 10^{-4}$, and have been determined by a numerical optimization. The shadowed region corresponds to the estimated visibility for state-of-the-art technology, $\lambda \simeq 60$–130. 2D plot (back): maximum visibility $\xi$ for different values of the parameter $\lambda'$. The blue circles have been evaluated numerically from the histograms in the 3D plot (and others). The error bars of the Monte Carlo simulation (black lines inside) have been determined assuming Poissonian statistics in each bin, and are negligible on this scale. The red dotted curve comes from our model for the visibility, and the black solid curve is the simplified expression presented in Eq. (10). We consider $\bar{N}_{\mathrm{eff}} = 1$. See Methods for more details. **b** Expected outcomes for $\lambda' = 75$ and $\bar{N}_{\mathrm{eff}}$ being 1 (full), 10 (dashed) and 100 (dotted line). The parameter $\Delta n_b$ has been optimized to achieve maximum visibility for each value of $\bar{N}_{\mathrm{eff}}$

As an outlook, it is desirable to extend this work to the optomechanical case, where mechanical systems with a similar quadratic coupling have recently been studied[33,45,46], but conditions to have a successful phonon QND measurement have not been yet determined. The electromechanical systems considered here can be described with Kirchoff's laws, which give rigorous results within a well-defined model. The physical mechanisms behind the heating are identified to be the Johnson–Nyquist noises associated to the resistors, and fabrication imperfections. For comparison, the exact description of dissipation in a multi-mode optomechanical system may be more involved. Nevertheless, the results presented here could be useful for guiding the intuition towards QND detection in the optical regime. As a further extension, it would be interesting to investigate the effect of squeezing. By reducing the vacuum noise, squeezing can lead to a direct improvement in $\lambda$, thus reducing the physical requirements for the QND detection.

## Methods

**The double-arm circuit.** The Hamiltonian for the system in Fig. 1c is given by

$$
\begin{aligned}
\hat{\mathcal{H}} = \hbar\omega_{\mathrm{m}}\hat{b}^\dagger\hat{b} + \frac{\hat{\Phi}_a^2}{4L} + \frac{\hat{Q}_a^2}{C_0} + \frac{\hat{\Phi}_s^2}{L+2L_0} + \frac{\hat{Q}_s^2}{4C_0} \\
+ \frac{g_1}{C_0\omega_s}\hat{Q}_a\hat{Q}_s(\hat{b}+\hat{b}^\dagger) + \frac{g_2}{C_0\omega_s}\hat{Q}_a^2\hat{b}^\dagger\hat{b} + \frac{g_2}{4C_0\omega_s}\hat{Q}_s^2\hat{b}^\dagger\hat{b},
\end{aligned}
\tag{11}
$$

where subscripts "a" and "s" indicate the asymmetric and the symmetric electrical fields, respectively. From Eq. (11) and using Kirchoff's laws, it is possible to determine the equations of motions, including noises and decays. The normalized distance $D^2 = d^2/\sigma^2$ is obtained assuming the phonon number to be constant within $T$—that is: setting $g_1 = 0$—so that the asymmetric and symmetric fields decouple. Looking at the phase quadrature of the reflected signal $\hat{V}_{\mathrm{out}} = \hat{V}_{\mathrm{in}} - \gamma_t\hat{\Phi}_s$, we determine $d$. The noise $\sigma$ is the sum of vacuum

noise from the input coherent field, and the Johnson–Nyquist noises of the resistors.

As discussed above, the heating $\Delta n_b = \langle\hat{n}_b(T)\rangle$ has two contributions: asymmetries leading to a non-vanishing linear coupling $g_r$, and the charge redistribution. The first is found by assuming $R \ll R_0$ and $L \ll L_0$, such that the circuit in Fig. 1c is equivalent to the one in Fig. 1b, for which we already know $\Delta n_b$. The contribution from charge redistribution is determined from the Hamiltonian in Eq. (11) neglecting the quadratic interaction, which does not alter the phonon number. The strongly driven symmetric electrical field is then substituted with its steady state, obtained assuming a constant photon flux. The time evolution of $\langle\hat{n}_b\rangle$ is finally found by looking at the equations of motion for the asymmetric field and the mechanical creation/annihilation operators. With the amplitude of the symmetric mode replaced by its steady state, these equations are now linear in the annihilation (creation) operators $\hat{b}$ ($\hat{b}^\dagger$) and can be solved by standard optomechanics techniques.

**Asymmetric circuit.** To obtain Fig. 3, we analyse the system in the presence of asymmetries. First, we derive the generalization of the Hamiltonian in Eq. (11) with unequal rest capacitors, resistors, inductors and linear couplings. Differently from above, we linearize the symmetric/asymmetric electrical fields around their mean values ($\hat{Q}_{a/s} \rightarrow \langle\hat{Q}_{a/s}\rangle + \delta Q_{a/s}$ and $\hat{\phi}_{a/s} \rightarrow \langle\hat{\phi}_{a/s}\rangle + \delta\phi_{a/s}$), and the mechanical creation/annihilation operators ($\hat{b}^{(\dagger)} \rightarrow \langle\hat{b}^{(\dagger)}\rangle + \delta b^{(\dagger)}$). Here, besides the usual oscillatory behaviour of the mechanical operators $\langle\hat{b}^{(\dagger)}\rangle$, the amplitude is generally time dependent[47]. This can be understood by looking at Eq. (11); since both the electrical fields have now non-zero average, the three body interaction $\propto \hat{Q}_a\hat{Q}_s(\hat{b}+\hat{b}^\dagger)$ is equivalent to a force directly driving the mechanical system. Once solutions for the averages are found, it is possible to determine the variations, and finally the time evolution of the phonon number.

**Optimization of the visibility.** To obtain Fig. 5 we rely on both an analytical and a numerical optimization of the visibility $\xi$. To determine the red, dotted curve, we assume that the initial mechanical state is thermal, such that the occupations of the Fock states can be found. Given $\Delta n_b$, the probability to jump once either up or down during the measurement time $T$ is a Poissonian process. The probability distribution function for the outcomes $V_M$ can then be obtained and maximized, by varying $\Delta n_b$. The histograms and the blue points are derived with Monte Carlo simulations, where the time evolution of single mechanical trajectories are replicated with the stochastic wave-function method[48]. Importantly, every measurement interval of duration $T$ has been discretized, to allow for multiple jumps. The parameter $\Delta n_b$ is then varied to find the best visibility $\xi$.

## Data availability

All material related to this work can be found at https://sid.erda.dk/share_redirect/eUaGoI8JbN.

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

## Acknowledgements

We gratefully acknowledge funding from the European Union Seventh Framework Programme through the ERC Grant QIOS, the European HOT network, and the Danish Council for Independent Research (DFF). We thank Emil Zeuthen and Albert Schliesser for fruitful discussions.

## Author contributions

O.K., A.S.S. and F.M. conceived the study. L.D. derived the main results, did the numerical calculations and wrote the first draft. All authors contributed to the manuscript. A.S.S. supervised the project.

## Additional information

**Competing interests:** The authors declare no competing interests.

