## [Peer Review File · Nature Communications]

Reviewers' comments:

Reviewer #1 (Remarks to the Author):

In this paper the authors analyze a new approach for performing a quantum non-demolition measurement of the phonon number of a nanomechanical resonator. Specifically they study the capacitive coupling of a higher order drum mode of a graphene membrane to a microwave LC resonator. In this case the linear coupling term disappears due to symmetry and ideally a purely quadratic optomechanical coupling is obtained. This setup then can be used to detect the phonon number of the membrane without introducing additional heating for the linear term, which is a major bottleneck in related schemes based on, e.g., a membrane-in-the-middle optomechanical setup. In their analysis the authors take residual linear couplings and dissipation through higher order electromagnetic modes into account and discuss a set of experimental parameters, which would allow a detection of single-phonon quantum jumps through the output of the driven LC resonator.

The implementation of a QDN measurement for phonons is one of the key goals in the field of optomechanics and new ideas to make this task experimentally more feasible are certainly of interest to the community. The paper is well written and the derivations seem to be correct. However, in the end the authors only compare the quadratic measurement strength with the resonator-induced linear heating rate and they assume rather extreme system parameters. Therefore, from the current discussion it is not clear, whether presented scheme is i) experimentally feasible, and ii) systematically better than previous schemes or if the improvements are merely based on the optimistic parameters assumed.

-) The current discussion creates the impression that the mechanical resonator modulates the whole capacitance of the circuit. If this is the case, a quick estimate shows that the system could directly reach the single photon strong coupling regime $g_1 > \gamma_t$, where also previous schemes for phonon QND measurements would work. To judge the relevance of this it would be very interesting for the reader to know the absolute values of all the coupling parameters, not only the ratios.

-) If a more realistic scenario is assumed the coupling of the graphene membrane will probably be very small, since it only modulates a tiny fraction of the total capacitance of the LC resonator. It is not clear that the resulting quadratic coupling is sufficiently strong to be observable. While this coupling will be amplified by the number of microwave photons N_{mw} , this number can not be arbitrarily large. For example, in a related setting [T. Rocheleau et al. arXiv:0907.3313] strong heating effects are observed for $N_{mw} > 10^8$. The assumed driving power of 15 μW corresponds to 10^{13} photons. Again, the absolute value of the coupling g_2 is very important to judge the feasibility of the presented scheme.

-) In the paper the authors assume that during the measurement the heating of the mechanical mode comes only from the linear coupling term, while the bare mechanical damping/rethermalization is neglected. Since no absolute values for the coupling constants are provided, it is not clear if this assumption is justified. If the cited value of $T=2.6ms$ for the measurement time is used, a quality factor of $Q > 10^7$ would be required to neglect the bare damping. What are the precise requirements on the mechanical quality factor and are the required quality factors realistic for a graphene sheet?

In summary, a clarification of these points is necessary to judge the feasibility and therefore also the relevance of this detection scheme. Since there are several other schemes to implement a phonon QND measurement (see e.g. Irish, Schwab, PRB 68, 155311 (2003) or H. Kaviani et al., Optica, 2, 271-274 (2015)), which under the same conditions might perform similar or even better, the current analysis is not fully convincing.

Reviewer #2 (Remarks to the Author):

The manuscript titled "quantum nondemolition measurement of mechanical motion quanta" proposes a type of device to read out quantum mechanical oscillators' occupation numbers directly. As it is well known to the researchers, the mechanical motions of a quantum oscillator is difficult to see, because usually something has to interact with it to get its information, thus interfering with it to change its motion. A feasible QND setup is certainly very meaningful to the progress in the relevant research fields. If the design of the current setup is sufficiently workable, the manuscript can be considered to publish in Nature Communications.

On the other hand, there are several points throughout the manuscript, which are quite uncertain to me. I would like to ask the authors to provide the corresponding clarifications regarding those points, before I can surely judge whether the work is publishable in the journal. The comments are as follows.

First of all, I have to admit that the supplementary material for the manuscript is much more readable than the main text. The current version of the main text looks like the style of Physical Review Letters. The authors may check if Nature Communications allows higher word limit. If allowed, many essential points in the supplementary material can be move to the main text to make the logic more coherent and text more readable. In a sense this is a work involving considerable technical issues, which are essential to the understanding of the paper. For example, a main issue of read-out phonon number by homodyne detection of V_m , as in sec. S1A, can be more detailed to let the readers clearly see the principle of the devices.

As indicated in the manuscript, the setup is used to measure the phonon number in cooled ground state of mechanical membrane (in Fig. S8 of supplementary material). It is meaningful indeed since the setup is to find the quantum feature close to ground state. But is it also interesting to consider a membrane at room temperature (in a state of thermal equilibrium with environment)? Then whether will there be big difference for the plots in Fig. 3?

Heating by measuring process must be well understood for the devices, since it should be insignificant to a QND. I would like to know more about several features in dealing with the phenomenon.

(1) The system Hamiltonian in (S17) is linearized by expanding around the finally averaged or stable value of charge/flux operator. Does that mean these quantities become stable very quickly? Fig. S8 show that the measurements are performed periodically. It means V_{in} signal could last shortly? Then how such pulse like drive can make the photon flux $\tilde{\alpha}$ in (S12) quickly stabilize? Or the measurement periods are always long enough? The α in (S18) is a time average over measurement period. Is it more reasonable to replace it with the time-dependent $\tilde{\alpha}$? The linearization of (S17) by means of expansion around (S18) is an approximation?

(2) There also exist two different linearizations for a similar system in optomechanics. One also expand the phonon operators around their average values (see PRA 77, 033804 (2008)). The other is to linearize to a time-dependent form (see PRL 118, 233604 (2017)). The authors should indicate the variations in linearization technique.

(3) In the "double arm" part, Sec. S2B, the linearization is around a time-dependent average (S38). As a result the differential equations carry oscillating coefficients. The corresponding equations in frequency domain, then, have the terms of displaced-frequency variables. Is the expression in (S42) an exact one to eliminate the displace terms like $b[\Omega_k - 2\omega_s]$, etc from (S41a)-(S41c)?

(4) The analytical expression (S61) is found with a linearization procedure neglecting the nonlinear effect proportional to g_2 . Meanwhile, some plots in Fig. S7 are claimed to be found with the full simulation with (S64), which includes the effect of g_2 . Indeed, Fig. S5 shows the sideband $\omega_m \pm 2\omega_s$ due to the nonlinear effect. However, the analytical results perfectly match the full simulation in Fig. S7. Does that mean the full simulation is also with the nonlinear effect of g_2 , and only include the fluctuations of R, C and L? The value of g_2 is not seen in Fig. S7. If so, the full simulation in Fig. S7 would be less non-trivial. Is it possible to add a considerable effect of g_2 to such simulation?

One assumption in the measurement part S3 is to confine only one jump of Fock states during measurement procedure. It could explain why the ground state instead of a state at finite temperature is considered here, since the latter is more likely to cause more complicated jumps. One question is whether the form of input $V_{in}(t)$ can affect such jump too. In the probability distribution function (S65), the contribution from the jump part is an integral over V value between two points. How to understand this expression since the jump is only a discrete one between the points $(i-1)d$ and id , while the contribution is from a continuous distribution between them?

Overall, it is an interesting work to tackle a challenging problem of QND of mechanical motion. The above mentioned treatments to justify a valid description should be, however, clarified further.

Reviewer #3 (Remarks to the Author):

This manuscript presents an electromechanical setup that could be capable of observing the quantized energy levels of a mechanical oscillator. A direct measurement of this represents a major milestone in the study of quantum mechanics. Although ideas for how to observe this in similar systems have been presented previously, the required parameter regime has precluded experiments from performing the measurement. Here, the authors present a scheme that appears feasible with existing technology, which I believe would be of great interest to the broader community. For that reason, I recommend acceptance of the manuscript with some revision to aid in the readability for a more general audience.

Specifically, I find that figure 2 should be critical in helping the reader understand the scheme being presented, but the explanation for the figure is minimal and scattered throughout the manuscript. I suggest that a more detailed discussion of this figure, with more discussion of the important features, would improve the paper's accessibility.

The final conclusion could also be more precise. The statement "We see that it is possible to observe signatures of the mechanical energy quantization above $\lambda \geq 32$..." is a hard to interpret. First, $\lambda=32$ is not shown in figure 3, and it is assumed that the reader can easily make the above conclusion. A little more discussion about how to interpret figure 3 would be helpful.

Reviewer #4 (Remarks to the Author):

This manuscript reports an approach to phonon number measurement in a mechanical oscillator using x -squared coupling in an electromechanical system. I think the results - if correct - are quite nice. The realization of phonon number measurement in a massive system would be an important fundamental milestone for quantum physics, and as the authors explain very clearly, has proved very challenging to achieve both because of the very weak necessary quadratic interactions inherent in such schemes and because deleterious linear coupling causes decoherence.

The authors emphasize that the original scheme to realize phonon counting - a membrane-in-the-middle approach using two coupled cavities - turns out to be fundamentally flawed and requires exquisitely high linear interaction strengths (at the single photon strong coupling level) to achieve. Their scheme is different in that it does not rely on two coupled cavities to generate a quadratic nonlinearity, but relies instead on the intrinsic quadratic nonlinearity of the system. Ideas along these lines have, for instance, been proposed for evanescently coupled this approach allows the requirement of single photon strong coupling to be removed since the linear and quadratic coupling strengths become independent.

From my perspective, given that the key concept in this paper is relatively straightforward to understand, I consider a significant part of the novelty of the paper to be in the proposed implementation with electromechanics. I note that the electromechanical scheme they propose has, in its essence, been proposed before in Phys. Rev. A 91, 033835 (2015). One very nice addition in this paper is the recognition and beautiful physical explanation of the heating coming from spatial charge fluctuations in the capacitor plate.

Overall, I'm not convinced that the paper is novel enough for Nature Communications.

I have some concerns about the theoretical approach in the paper and the assumptions made that raise questions about some of the claims made. I would note that these issues, while I think important, were not primary in my consideration of the novelty of the work.

The main concern is that, in the main text, the authors neglect phonon jumps caused by the environment, assuming that the jumps caused by the linear optomechanical coupling dominate. This leads them to the conclusion that their scheme is different than previous schemes because it is only sensitive to the ratio g_2/g_1 . Of course, this is not strictly correct, because at low enough interaction strengths, the required measurement duration will push one into regimes where environmental jumps are significant. Compounding this issue, the authors assume that the mechanical oscillator is in its ground state when calculating the jump probability. This is the best case scenario, and indeed, is not sufficient for phonon counting - one must be able to determine whether the oscillator is in one or the other of two phonon number states, so the jump rate should be calculated at the very least for both $|0\rangle$ and $|1\rangle$. There is a fundamental difference for $|1\rangle$, in that the phonon number can decay. This leads to a fundamentally different looking jump probability with a term that doesn't depend on the mean phonon number in the environment. This is a critical point, I believe, because you cannot simply go to a very low temperature to freeze this term out. As a result, I expect that, analyzing the single phonon case will lead to stringent conditions on the mechanical decay rate that are currently absent from the paper.

As a much more minor point, I struggle to follow the authors justification for neglecting the $b\dot{b}$ and $\dot{b}b$ terms in Eq (2b) and other places. I understand that the dynamics from these terms will average out in the measurement and heating, but it's not clear to me that they will not affect the coherent dynamics of the system in a meaningful way. Some stronger justification would be appreciated here.

Reviewer 1:

We thank Reviewer 1 for saying that “the paper is well written and the derivation seems to be correct”, and for his/her overall positive comments.

The answer to the questions asked by Reviewer 1 are now included in the main text. Furthermore, we analysed them in details in the last section of the revised supplemental material (S4), where all the parameters of the considered setup are given/derived in details. In the following, we give detailed responses to the concerns of Reviewer 1.

- 1) *“The current discussion creates the impression that the mechanical resonator modulates the whole capacitance of the circuit. If this is the case, a quick estimate shows that the system could directly reach the single photon strong coupling regime $g_1 > \gamma_t$, where also*

previous schemes for phonon QND measurements would work. To judge the relevance of this it would be very interesting for the reader to know the absolute values of all the coupling parameters, not only the ratios.”

The Reviewer is right; if there were no stray capacitance C_s , the linear coupling g_1 would be $\sim(2\pi)700$ kHz, which is bigger than the electrical damping $\gamma_t = (2\pi)150$ kHz, and the system would be in the strong coupling regime. As noted by the Reviewer, however, there will in practice always be a significant stray capacitance C_s . The presence of such stray capacitance necessarily limits the coupling strength, and our previous estimates applied to a more realistic situation with large stray capacitance. To have strong coupling, we would require C_s to be at most $C_s = 3.8 C_0$, while a stray capacitance that is hundreds or even thousands of times bigger than C_0 is expected. Assuming $C_s = 100 C_0$ [similar to X. Songet al., Nano Letters 12, 198 (2012)], the linear coupling g_1 becomes $\sim(2\pi)7$ kHz, and the quadratic $g_2 \sim(2\pi)1$ Hz. With these values the system is far from the strong coupling regime, but nevertheless is still able to show the quantization of the energy. We agree with the Reviewer that due to the compressed nature of our first manuscript, the discussion of this issue was not sufficiently detailed. In the revised version we therefore give a much more thorough discussion of the relevant coupling strengths and the issue of stray capacitance.

- 2) *“If a more realistic scenario is assumed the coupling of the graphene membrane will probably be very small, since it only modulates a tiny fraction of the total capacitance of the LC resonator. It is not clear that the resulting quadratic coupling is sufficiently strong to be observable. While this coupling will be amplified by the number of microwave photons N_{mw} , this number can not be arbitrarily large. For example, in a related setting [T. Rocheleau et al. arXiv:0907.3313] strong heating effects are observed for $N_{mw} > 10^8$. The assumed driving power of $15 \mu W$ corresponds to 10^{13} photons. Again, the absolute value of the coupling g_2 is very important to judge the feasibility of the presented scheme.”*

We agree with the Reviewer that stray capacitance will limit the coupling strength (as discussed above) and that there is a limit to how many photons one can use. The photon flux sent into the circuit is strongly dependent on two parameters: the mechanical quality factor (see below) and the absolute value of the coupling. The latter is closely related to point 1) above. In the revised manuscript we now give a detailed discussion of these dependencies and calculate the required photon numbers for various stray capacitances and quality factors. In particular, we show that with a stray capacitance being 100 times the bare one (implying $g_2 = (2\pi)1$ Hz) and a mechanical quality factor between 10^5 and 10^7 , the required number of microwave photons in the cavity N_{mw} can vary from 10^{10} to 10^8 . We believe, however, that compared to the work quoted by the referee the heating may be much reduced for our system. The experimental results indicate that the heating is due to electric fluctuations. Hence, the heating will be suppressed by the vanishing linear coupling g_1 due to symmetry.

- 3) *“In the paper the authors assume that during the measurement the heating of the mechanical mode comes only from the linear coupling term, while the bare mechanical damping/rethermalization is neglected. Since no absolute values for the coupling constants are provided, it is not clear if this assumption is justified. If the cited value of $T=2.6ms$ for the measurement time is used, a quality factor of $Q>10^7$ would be required to neglect the bare damping. What are the precise requirements on the mechanical quality factor and are the required quality factors realistic for a graphene sheet?”*

As noted by the Reviewer, whether or not the mechanical damping can be ignored depends on the time scale of the experiment. By increasing the flux of photons the measurement time can be decreased, which reduces the required quality factor. This question is however then linked to the number of photons in the cavity (point 2) above). In the revised version we now give a detailed discussion of the proposed experimental conditions and how the quality factor influence the photon number. The mechanical quality factors of graphene sheets at cryogenic temperatures are usually between 10^5 and 10^6 [see for example V. Singh et al., Nat Nano 9, 820 (2014), or P. Weber et al., *Nano letters* 14.5 (2014): 2854-2860.]. In the manuscript we consider values in the range from 10^5 to an optimistic 10^7 , and show that the remaining experimental parameters can be adjusted to compensate a decrease in the mechanical quality factor. As an example, for $Q = 10^6$ and assuming that *both* the mechanical and the electrically induced reservoirs are at an effective temperature of 14 mK, the measurement time is 0.1 ms. Changing Q to 10^4 , the time becomes $1 \mu\text{s}$, to compensate for the increase in the mechanical heating. The required average intracavity photon number is now increased to about 10^{11} , but even in this case, we may suppress the effect of heating if we run the experiment in a pulsed regime.

Reviewer 2:

We would like to thank Reviewer 2 for saying that “the manuscript can be considered to publish in Nature Communications”, if his/her questions are answered with sufficient details. We hope that the extended version of the manuscript can convince him/her on the feasibility of the proposal, and that the whole text is more coherent and readable. In the following, we answer to the questions asked in order to clarify all the points of Reviewer 2.

- 1) *“As indicated in the manuscript, the setup is used to measure the phonon number in cooled ground state of mechanical membrane (in Fig. S8 of supplementary material). It is meaningful indeed since the setup is to find the quantum feature close to ground state. But is it also interesting to consider a membrane at room temperature (in a state of thermal equilibrium with environment)? Then whether will there be big difference for the plots in Fig. 3?”*

Literally doing the experiment at room temperature would be challenging for the considered experimental setup, since high quality LC resonators require superconducting circuits. But indeed it is also interesting to study the system at higher temperatures. This would have two effects. 1) it would influence the value of λ through n_e in Eqs. (S36, S62), and 2) it would lead to a higher mean quantum number. The simulation in the old Fig. 3 (now Fig. 5), was already performed in a state of thermal equilibrium with the environment, although we assumed cryogenic cooling (and/or an additional MW cooling) to reach rather low temperatures corresponding to an average of one excitation. Reducing the cooling results in a higher mean occupation, but actually leads to a slightly better visibility for the part of the signal which is near the ground state [see Eq. (7)], since it increases the relative probability to be in the first excited state (as compared to the ground, see Fig. 5(b)). On other hand, running the experiment with an average number $n_m \gg 1$ would mean that the probability to be in the ground state is $1/n_m$. Hence the required time to gather sufficient statistics to see distinct peaks would grow as n_m . In the revised version we give a more detailed discussion of the exact experimental condition assumed for Fig. 5, which also clarifies this issue. Furthermore, Fig. 5 also shows how the results would look at an elevated mean number of phonons.

- 2) *“The system Hamiltonian in (S17) is linearized by expanding around the finally averaged or stable value of charge/flux operator. Does that mean these quantities become stable very quickly? Fig. S8 show that the measurements are performed periodically. It means V_{in} signal could last shortly? Then how such pulse like drive can make the photon flux $\tilde{\alpha}$ in (S12) quickly stabilize? Or the measurement periods are always long enough? The α in (S18) is a time average over measurement period. Is it more reasonable to replace it with the time-dependent $\tilde{\alpha}$? The linearization of (S17) by means of expansion around (S18) is an approximation?”*

There are different ways in which this experiment could be performed. In the revised version we explain that the main considered scenario is one in which all fields are incident and measured continuously. In this situation the approximation is fully justified by just letting the experiment run for sufficiently long time. The grouping into distinct measurement periods is then done by simply averaging the recorded signal over a period T . To avoid confusion about this point we have removed the blank spaces in Fig. S8, which we believe may have been misleading.

The measurement could also be done in a more pulsed manner. In this case the description should ideally include a time dependent mean field, as noted by the Reviewer. However, the measurement time is hundreds of times longer than the characteristic lifetime of the microwave cavity γ_t^{-1} . We thus believe that using the time independent steady state of the strongly driven microwave field is a good approximation even with pulsed probing. In the revised version we remark, however, that this is an approximation and that if rapidly varying fields are used, this step should be replaced by suitable time dependent fields. Since such a change will predominantly add a time dependent mean value to the output, this will not affect the final conclusion, provided that different shots of the experiments are sufficiently reproducible, such that they add the same mean value.

In the linearization of Eq. (S17) to arrive at Eq. (S19) we have neglected terms that are quadratic in $\delta\alpha$ and its conjugate. This can be done, since they are not enhanced by the strong field α , and are rapidly oscillating at twice the electrical frequency. In the revised version we explicitly mention this after Eq. (S19).

- 3) *“There also exist two different linearizations for a similar system in optomechanics. One also expand the phonon operators around their average values (see PRA 77, 033804 (2008)). The other is to linearize to a time-dependent form (see PRL 118, 233604 (2017)). The authors should indicate the variations in linearization technique.”*

In general, for deriving the analytical results presented in our work, we always linearize the strongly driven electric field operators, but do not linearize the asymmetric electrical field and the phonon operators, since there is only some perturbative dynamics affecting them. However, all operators have been linearized for solving the system in Eq. (S65). In that case, a linearization of the phonon creation/annihilation operators is required, since they are directly driven by the steady states of the two electrical fields (not only one). Then, we solve the system assuming that b and its conjugate start in some initial value (in general 0, but can be freely chosen), and evolve them according to the dynamics induced by the electrical subsystems. We consider their average values first, and their small variations later. The average values are, in this case, time dependent, such that the technique we employed is more similar to the one presented in PRL 118, 233604 (2017). This has been now clarified in the supplementary information.

- 4) *“In the ‘‘double arm’’ part, Sec. S2B, the linearization is around a time-dependent average (S38). As a result the differential equations carry oscillating coefficients. The corresponding equations in frequency domain, then, have the terms of displaced-frequency variables. Is the expression in (S42) an exact one to eliminate the displaced terms like $b[\Omega_k - 2\omega_s]$, etc from (S41a)-(S41c)?”*

In the equation (S42) (now (S43)) we neglected off-resonant terms such as $b[\omega_m \pm 2\omega_s]$, and in general we only kept the mechanical operators resonant with the mechanical frequency ω_m . This approximation has been investigated in Fig. S6, where these off-resonant terms are taken into account, and proven to be negligible. We comment below Eq. (S44) that we neglect such off-resonant terms.

- 5) *“The analytical expression (S61) is found with a linearization procedure neglecting the nonlinear effect proportional to g_2 . Meanwhile, some plots in Fig. S7 are claimed to be found with the full simulation with (S64), which includes the effect of g_2 . Indeed, Fig.S5 shows the sideband $\omega_m \pm 2\omega_s$ due to the nonlinear effect. However, the analytical results perfectly match the full simulation in Fig. S7. Does that mean the full simulation is also with the nonlinear effect of g_2 , and only include the fluctuations of R, C and L? The value of g_2 is not seen in Fig. S7. If so, the full simulation in Fig. S7 would be less non-trivial. Is it possible to add a considerable effect of g_2 to such simulation?”*

The figure S5 in the supplementary is indicating the off-resonant contributions that come from the linearization of the electrical conjugate momenta, and not from the quadratic coupling (please see point 4 above). The quadratic coupling g_2 has been ignored for drawing figures S6 and S7. This is eligible since the main effect of g_2 is that it introduces a term in the Hamiltonian that is proportional to the phonon number n_b , and therefore does not influence the time evolution of n_b . Notice that the structure of the Markovian baths results in an exponential decay of the mechanical state, implying that the quantum Zeno effect does not influence the system’s dynamics, thus justifying an independent treatment of the heating and measurement, despite the system being subject to a strong measurement. The quadratic coupling is derived from the second order expansion of the position operator, and thus also carries terms such as $\hat{b}\hat{b}$ and its conjugate (which do not commute with the phonon number). These terms are however strongly suppressed by the rotating wave approximation since they are far off resonant. By using Fermi golden rule, we estimate the rate at which these two phonon processes happen $\sim \frac{g_2^2 \alpha^2 \gamma t}{4\omega_m^2}$. With the parameters introduced in section S4 of the supplemental material, we can estimate this quantity to be 4 orders of magnitude smaller than the rate relative to single phonon processes, $\Delta n_b / T$. In essence, this difference comes from the fact that g_1 is just much bigger than g_2 . The only reason why g_2 is relevant is that it has a resonant contribution. When considering off-resonant effects (sidebands) the terms involving g_1 are much more important. In the revised version we explicitly state that figure S7 is derived neglecting g_2 . Whether or not it is in principle possible to include the g_2 contribution in this simulation is an interesting question, but this is beyond the scope of the present work.

- 6) *“One assumption in the measurement part S3 is to confine only one jump of Fock states during measurement procedure. It could explain why the ground state instead of a state at finite temperature is considered here, since the latter is more likely to cause more complicated jumps. One question is whether the form of input $V_{in}(t)$ can affect such jump too. In the*

probability distribution function (S65), the contribution from the jump part is an integral over V value between two points. How to understand this expression since the jump is only a discrete one between the points $(i-1)d$ and id , while the contribution is from a continuous distribution between them?"

As noted above and more carefully explained in the revised version, in our proposal for an experiment (Fig. S8), the membrane always starts in a thermal state (in principle not necessarily of low occupation number). We mainly look at the outcomes corresponding to the first few Fock states because this is where we have the largest visibility. In section S3 of the supplemental, we do not in general assume that only a single jump happens. We first make a simplified model where we indeed assume a single jump, and then numerically simulate what happens when we allow for multiple jumps. For the QND measurement we need to have a small overall probability to jump within each measurement. This also implies that there is very little chance that two- or more jumps happen in the regime where the QND probing is efficient. This is the reason why our simplified model gives good agreement with the simulation in the regime of high λ in Fig. 5. For small values of λ , on the other hand the two descriptions slightly deviate because of the possibility of multiple jumps (see the first numerical points in Fig. 5).

Eq. (S66) (old Eq. (S65)) represents the probability distribution function of an outcome V_m . V_m is the time average of the phase of the reflected signal, and in Eq. (S13a), the integral is inside α , and is not explicitly written because the flux α/T is assumed constant. Eq. (S66) has two terms in the sum describing, respectively, all the cases in which the mechanical state did not jump, and all the ones in which it did. The reason for integrating the signal is that even if there is only one jump, that jump is equally probable to happen at any time within the measurement. Since here we only consider time independent photon fluxes, the time dependence is not relevant. We hope that this issue is clearer in the revised version with the more careful explanation of our considered experimental setup.

Reviewer 3

We thank Reviewer 3 for his/her very positive comments, in particular recommending acceptance of the manuscript.

- 1) *Specifically, I find that figure 2 should be critical in helping the reader understand the scheme being presented, but the explanation for the figure is minimal and scattered throughout the manuscript. I suggest that a more detailed discussion of this figure, with more discussion of the important features, would improve the the paper's accessibility.*

We agree that in the first version of the manuscript, figure 2 wasn't described appropriately in the main text. Now we extended it, including more detailed descriptions on what it means, and which implications it has.

- 2) *The final conclusion could also be more precise. The statement "We see that it is possible to observe signatures of the mechanical energy quantization above $\lambda \geq 32$..." is a hard to interpret. First, $\lambda=32$ is not shown in figure 3, and it is assumed that the reader can easily make the above conclusion. A little more discussion about how to interpret figure 3 would be helpful.*

The limit for λ we have identified is very much qualitatively, and is used as a reference to know in advance whether an experimental setup would or would not achieve the QND measurement. We have modified the section to make it clearer and remove the ambiguity. The value we found corresponds to a visibility of 20%, which we believe to be a relatively clear signature of the quantization of the energy. As it is possible to see from Fig. S9 in the supplementary, already at $\lambda = 20$ some signatures are visible. In the revised version we quote a less precise value of approximately 40.

Reviewer 4

This manuscript reports an approach to phonon number measurement in a mechanical oscillator using x -squared coupling in an electromechanical system. I think the results - if correct - are quite nice. The realization of phonon number measurement in a massive system would be an important fundamental milestone for quantum physics, and as the authors explain very clearly, has proved very challenging to achieve both because of the very weak necessary quadratic interactions inherent in such schemes and because deleterious linear coupling causes decoherence.

The authors emphasize that the original scheme to realize phonon counting - a membrane-in-the-middle approach using two coupled cavities - turns out to be fundamentally flawed and requires exquisitely high linear interaction strengths (at the single photon strong coupling level) to achieve. Their scheme is different in that it does not rely on two coupled cavities to generate a quadratic nonlinearity, but relies instead on the intrinsic quadratic nonlinearity of the system. Ideas along these lines have, for instance, been proposed for evanescently coupled this approach allows the requirement of single photon strong coupling to be removed since the linear and quadratic coupling strengths become independent.

From my perspective, given that the key concept in this paper is relatively straightforward to understand, I consider a significant part of the novelty of the paper to be in the proposed implementation with electromechanics. I note that the electromechanical scheme they propose has, in its essence, been proposed before in Phys. Rev. A 91, 033835 (2015). One very nice addition in this paper is the recognition and beautiful physical explanation of the heating coming from spatial charge fluctuations in the capacitor plate.

We thank Reviewer 4 for carefully reading the manuscript, and his/her appreciation of the result and the analysis of the charge redistribution in the system. We carefully consider his/her comments, and are sorry to discover that he/she does not think that our work is novel enough. We agree that the idea behind, which relies on eliminating the linear coupling with symmetry, has been proposed before, for instance in the optomechanical setups that demonstrated quadratic coupling, but for which strong (linear) coupling is required for achieving QND detection. A related idea also appears in the paper quoted by the referee [Phys. Rev. A 91, 033835 (2015)], where symmetry is used to have a quadratic coupling in an electric setup, thereby essentially simulating the mechanical model. However, such simulator does not contain a mechanical degree of freedom, and the non-linear interaction comes from SQUIDs that, themselves, have huge nonlinearities (MW photon QND measurement has been already achieved with Josephson junctions). Such a setup thus cannot be used to prove the quantization of mechanical energy. We also note that this paper neglects similar coupling terms as the ones which were shown to be the major obstacle to the QND detection. This paper thus sheds no light on the central issue of whether this is justified and under which conditions one can achieve QND detection. To our knowledge, our work is the first that clearly demonstrates feasibility of phonon QND

measurement for currently available technology, and actual mechanical resonators. And this is demonstrated considering all sources of heating – even the ones usually neglected – and all experimental steps, from the design till the probe. Even though the idea of using symmetry and quadratic couplings has appeared before, we believe that our work is the first to identify the exact conditions under which QND detection is possible using this approach. In our opinion this is an important contribution that goes beyond the specific implementation that we propose. This insight will be important to anybody trying to achieve QND detection with different implementations. In the revised version we highlight this aspect of our work.

In the following, we answer to the question that Reviewer 4 made.

- 1) *“The main concern is that, in the main text, the authors neglect phonon jumps caused by the environment, assuming that the jumps caused by the linear optomechanical coupling dominate. This leads them to the conclusion that their scheme is different than previous schemes because it is only sensitive to the ratio g_2/g_1 . Of course, this is not strictly correct, because at low enough interaction strengths, the required measurement duration will push one into regimes where environmental jumps are significant.”*

As already mentioned (please see answer 3) in the response to Reviewer 1), we do not neglect phonon jumps induced by the environment, see e.g. Eq. (S62). However, we agree that there was a lack of details about the mechanical reservoir in the first version of our manuscript. This has now been clarified both in the main text and in the supplementary. In short, the idea is that high power and short measurement time allow us to neglect the mechanical reservoir (in principle, we can always increase the probing power, to neglect the environment). In the revised version we now give a detailed discussion of the influence of the environmental heating of the membrane.

- 2) *“Compounding this issue, the authors assume that the mechanical oscillator is in its ground state when calculating the jump probability. This is the best case scenario, and indeed, is not sufficient for phonon counting - one must be able to determine whether the oscillator is in one or the other of two phonon number states, so the jump rate should be calculated at the very least for both $|0\rangle$ and $|1\rangle$. There is a fundamental difference for $|1\rangle$, in that the phonon number can decay.”*

The quantity Δn_b and the rate $\Delta n_b/T$ are defined for a membrane being initially in its ground state. However, as explained in section S3 of the supplemental, from those it is possible to determine the rates at which any other state jumps down/up. In fact, as we more carefully explain in the revised version, we consider an initial mechanical thermal state, and not only the ground state. The rate $\Delta n_b/T$ is a universal quantity that does not only describe the system being in the ground state. In the revised version we explicitly state that this is only how we define Δn_b and that any other jump probability can be derived from it.

- 3) *“As a much more minor point, I struggle to follow the authors justification for neglecting the bb and $b^\dagger b^\dagger$ terms in Eq (2b) and other places. I understand that the dynamics from these terms will average out in the measurement and heating, but it's not clear to me that they will not affect the coherent dynamics of the system in a meaningful way. Some stronger justification would be appreciated here.”*

Regarding the terms $\hat{b}\hat{b}$ and its conjugate, we refer to the point 5) of our answer to Reviewer 2. Here it is important to remember that g_2 is in fact extremely small. The only reason why it is important

to include it, is that it has a resonant contribution. Our comparison is conducted when this resonant contribution is comparable to the linear sideband leading to heating, which are suppressed by being off-resonant. The off-resonant contribution of $\hat{b}\hat{b}$ and its conjugate are much smaller than the heating we consider, simply because g_2 is much smaller than g_1 . Furthermore, any sideband contribution (both from the linear and quadratic coupling) in the output signal is shifted in frequency. This means that it doesn't show up in a homodyne detection at the same frequency as the input (it disappears when integrating the photocurrent). In the revised version we explicitly state that we can ignore g_2 when calculating the heating since it is small compared to g_1 and that we can neglect the sidebands during the measurement since homodyne detection is frequency selective.

REVIEWERS' COMMENTS:

Reviewer #1 (Remarks to the Author):

In the revised version, the authors have substantially improved the discussions related to the physical implementation of their scheme and added many new estimates and actual numbers, which I had requested in my first report. From the current discussion it is now more clear, where experimental challenges lie and that no basic source of dissipation has been omitted. I think that also the concerns from the other reviewers have been adequately addressed in the replies and in the manuscript and the authors in particular clarified the novelty with respect to previous works. I have further complaints and consider the paper now suited for publication in Nature Communications.

Reviewer #2 (Remarks to the Author):

I have checked the authors' replies to the comments and questions in the last report. Their replies are satisfactory to clarify the technical issues. In particular, the current version of the manuscript has been significantly improved to a style of Nature Communications. So I recommend the publication of the manuscript in the current form.

Reviewer #4 (Remarks to the Author):

I would thank the authors for their thoughtful responses to my, and the other referees comments/concerns. Overall, however, I stand by my initial opinion that, while interesting, this work is just not of sufficient novelty to warrant publication in Nature Communications. The essential issue is that the concept of using symmetry to suppress linear optomechanical coupling and therefore achieve quantum nondemolition measurement of a mechanical resonator through the quadratic coupling without requiring the single photon strong coupling regime is well established already. This being the case, the main novelty of this manuscript is a specific proposal of an experiment that may (with ambitious but perhaps realizable assumptions) achieve this.

To give some more substance to my concerns here, some previous papers (both experimental and theory, and non-exhaustive) that discuss/demonstrate the suppression of the linear interaction term in the context of phonon number quantum nondemolition measurement (and therefore quantum nondemolition outside of the strong linear coupling regime) are:

Nature Communications 8 16024 (2014) (Experimental)
Nature Physics 5 909 (2009) (Experimental)
Physical Review A 85 053832 (2012)

While, some previous papers that point to pathways to experimentally feasible but challenging phonon number quantum nondemolition measurements are:

Optics Express 20 24394 (2012) (Experimental proposal)
Physical Review Letters 112 203603 (2014) (new theory)
Physical Review X 5 041024 (2015) (experimental proposal)
New Journal of Physics 16 055008 (2014)

It can be observed that, generally, proposals analogous in my view to this manuscript are found in

more technical journals.

A further comment I would make is that I find the manuscript to be - while generally clearly and well written - a bit misleading in places.

For example, the manuscript reads as if the idea of using symmetry to suppress the linear coupling and therefore avoid the strong coupling requirement is a new one in this manuscript, whereas this idea has existed for many years (see examples above). For instance, the statement in the abstract that "For optomechanical systems, the coupling to the environment was shown to preclude the detection of the mechanical mode occupation, unless strong single photon coupling is achieved." is false as should be clear from the references above that discuss suppression the linear coupling term. While the similar statement in the introduction referencing papers that must operate in the single photon strong coupling regime, but omitting those that don't have this requirement is misleading. Similar statements can be found throughout the text which very strongly imply that the "optomechanical case" requires strong coupling, whereas in fact this is only correct for a limited class of optomechanical configurations.

In the second round of reviewing, three Reviewers were consulted. Two are positive (1,2) and suggest publication in Nature communication with no further comments, and one (4) does not. We would like to sincerely thank Reviewers 1 and 2 for their response. We answer Reviewer 4's criticism in the following.

Reviewer 4:

We are very sorry to read that Reviewer 4 still believes that our work is not novel enough. At this point, we do not think that we are able to change his/her opinion, but still give point— to—point answers to his/her comments. In particular, we consider the claim according to which

“The essential issue is that the concept of using symmetry to suppress linear optomechanical coupling and therefore achieve quantum nondemolition measurement of a mechanical resonator through the quadratic coupling without requiring the single photon strong coupling regime is well established already. This being the case, the main novelty of this manuscript is a specific a proposal of an experiment that may (with ambitious but perhaps realizable assumptions) achieve this.”

To some extent we agree with this comment, but also do not think that this affects the novelty of our work. As already mentioned in the first round of reviews, we acknowledge that the use of symmetry to suppress the linear coupling is not a novel contribution of our work. Indeed it has been suggested to use symmetry in several articles. These previous suggestions fall into two categories: 1) Articles that relies on an induced quadratic coupling and 2) setups that exploit the intrinsic quadratic coupling. For the first category it was shown that strong single photon coupling is required. For the second only limited analysis has been performed. In addition to proposing a specific experiment (as acknowledged by the referee), we believe that our paper is also the first to precisely work out the conditions for when the second approach is possible.

Reviewer 4 cites several articles to support his words. Most of these articles [Nature Communications 8 16024 (2014), Optics Express 20 24394 (2012), Physical Review Letters 112 203603 (2014), Physical Review X 5 041024 (2015), New Journal of Physics 16 055008 (2014)] fall into the first category and aim to achieve the QND measurement by increasing the bare linear optomechanical coupling. This is exactly what we want to avoid, by using an intrinsic quadratic coupling, instead of an induced one. A few of the articles fall into the second category and propose experiments where an intrinsic quadratic coupling can be achieved [Physical Review A 85 053832 (2012), Nature Physics 5 909 (2009)]. These do not, however, perform any analysis on the feasibility of a phonon QND measurement, which is our main contribution to the field in general (apart from the specific experiment).

After reading the comments of the reviewer (including the point below), we realized that we did not give sufficient references to papers, which use the intrinsic quadratic coupling for optomechanical systems. We have therefore included the papers cited by the reviewer, which fall into this category. On the other hand, we do not feel it is so relevant to include the

references, which works by an increased bare linear coupling, since these are extensions of the approach we are trying avoid (where we cite the first papers to consider this possibility).

Another point raised by the Reviewer is that:

“For instance, the statement in the abstract that “For optomechanical systems, the coupling to the environment was shown to preclude the detection of the mechanical mode occupation, unless strong single photon coupling is achieved.” is false as should be clear from the references above that discuss suppression the linear coupling term. While the similar statement in the introduction referencing papers that must operate in the single photon strong coupling regime, but omitting those that don't have this requirement is misleading. Similar statements can be found throughout the text which very strongly imply that the “optomechanical case” requires strong coupling, whereas in fact this is only correct for a limited class of optomechanical configurations.”

We agree with the referee that our wording was slightly imprecise. As noted above there are two categories of approaches to QND detection in the optomechanical regime: 1) induced quadratic coupling and 2) direct quadratic coupling. The first category is by far the most studied in the field, and therefore our discussion focussed on this situation, where strong single photon coupling is required. In the revised version we have changed the text in order to be more precise, and included references to articles that study the intrinsic quadratic coupling. We now limit ourselves to noting that the usual approach to phonon QND measurement is limited to the strong coupling regime. But we agree that this does not imply that it is impossible to succeed in the phonon QND measurement using the intrinsic quadratic coupling. In fact, as we say in the conclusions of our manuscript, it is desirable to extend the work to the optomechanical case. This is something we are finalizing at the moment.